# Structured State Space Models for In-Context Reinforcement Learning

**Chris Lu**[*]
FLAIR, University of Oxford

**Yannick Schroecker**
DeepMind

**Albert Gu**
DeepMind

**Emilio Parisotto**
DeepMind

**Jakob Foerster**
FLAIR, University of Oxford

**Satinder Singh**
DeepMind

**Feryal Behbahani**
DeepMind

## Abstract

*Structured state space sequence* (S4) models have recently achieved state-of-the-art performance on long-range sequence modeling tasks. These models also have fast inference speeds and parallelisable training, making them potentially useful in many reinforcement learning settings. We propose a modification to a variant of S4 that enables us to initialise and reset the hidden state in parallel, allowing us to tackle reinforcement learning tasks. We show that our modified architecture runs asymptotically faster than Transformers in sequence length and performs better than RNN's on a simple memory-based task. We evaluate our modified architecture on a set of partially-observable environments and find that, in practice, our model outperforms RNN's while also running over five times faster. Then, by leveraging the model's ability to handle long-range sequences, we achieve strong performance on a challenging meta-learning task in which the agent is given a randomly-sampled continuous control environment, combined with a randomly-sampled linear projection of the environment's observations and actions. Furthermore, we show the resulting model can adapt to out-of-distribution held-out tasks. Overall, the results presented in this paper show that structured state space models are fast and performant for in-context reinforcement learning tasks. We provide code at `https://github.com/luchris429/s5rl`.

## 1 Introduction

Structured state space sequence (S4) models [12] and their variants such as S5 [38] have recently achieved impressive results in long-range sequence modelling tasks, far outperforming other popular sequence models such as the Transformer [42] and LSTM [16] on the Long-Range Arena benchmark [41]. Notably, S4 was the first architecture to achieve a non-trivial result on the difficult Path-X task, which requires the ability to handle extremely long-range dependencies of lengths $16k$.

Furthermore, S4 models display a number of desirable properties that are not directly tested by raw performance benchmarks. Unlike transformers, for which the per step runtime usually scales quadratically with the sequence length, S4 models have highly-scalable inference runtime performance, asymptotically using *constant* memory and time per step with respect to the sequence length. While LSTMs and other RNNs also have this property, empirically, S4 models are far more performant while also being *parallelisable across the sequence dimension* during training.

While inference-time is normally not included when evaluating on sequence modelling benchmarks, it has a large impact on the scalability and wallclock-time for reinforcement learning (RL) because

---

[*]Work done during an internship at DeepMind. Contact: christopher.lu@exeter.ox.ac.uk

37th Conference on Neural Information Processing Systems (NeurIPS 2023).

| | Inference | Training | Parallel | Variable Lengths | bsuite Score |
|---|---|---|---|---|---|
| RNNs | **O**(1) | **O**($L$) | No | **Yes** | No |
| Transformers | O($L^2$) | O($L^2$) | **Yes** | **Yes** | **Yes** |
| S5 with • | **O**(1) | **O**($L$) | **Yes** | No | N/A |
| S5 with ⊕ | **O**(1) | **O**($L$) | **Yes** | **Yes** | **Yes** |

Table 1: The different properties of the different architectures. The asymptotic runtimes are in terms of the sequence length $L$ assume a constant hidden size. The bsuite scores correspond to whether or not they achieve a perfect score in the median runs on the bsuite memory length environment.

the agent uses inference to collect data from the environment. Thus, transformers usually have poor runtime performance in reinforcement learning [33]. While transformers have become the default architecture for many supervised sequence-modelling tasks [42], RNNs are still widely-used for memory-based RL tasks [29].

The ability to efficiently model contexts that are orders of magnitude larger may enable new possibilities in RL. This is particularly applicable in meta-reinforcement learning (Meta-RL), in which the agent is trained to adapt across *multiple* environment episodes. One approach to Meta-RL, RL$^2$ [8, 43], uses sequence models to directly learn across these episodes, which can often result in trajectories that are thousands of steps long. Most instances of RL$^2$ approaches, however, are limited to narrow task distributions and short adaptation horizons because of their limited effective memory length and slow training speeds.

Unfortunately, simply applying S4 models to reinforcement learning is challenging. This is because the most popular training paradigm in on-policy RL with multiple actors involves collecting fixed-length environment trajectories, which often cross episode boundaries. RNNs handle episode boundaries by resetting the hidden state at those transitions when performing backpropagation through time. Unlike RNNs, S4 models cannot simply reset their hidden states within the sequence because they train using a fixed convolution kernel instead of using backpropagation through time.

A recent modification to S4, called Simplified Structured State Space Sequence Models (S5), replaces this convolution with a *parallel scan* operation [38], which we describe in Section 2. In this paper, we propose a modification to S5 that enables resetting its hidden state within a trajectory during the training phase, which in turn allows practitioners to simply replace RNNs with S5 layers in existing frameworks. We then demonstrate S5's performance and runtime properties on the simple bsuite memory-length task [32], showing that S5 achieves a higher score than RNNs while also being nearly two times faster when using their provided baseline algorithm. We also re-implement and open source the recently-proposed Partially Observable Process Gym (POPGym) suite [27] in pure JAX, resulting in end-to-end evaluation speedups of over $30x$. When evaluating our architecture on this suite, we show that S5 outperforms GRU's while also running over six times faster, achieving state-of-the-art results on the "Repeat Hard" task, which all other architectures previously struggled to solve. We further show that the modified S5 architecture can tackle a long-context partially-observed Meta-RL task with episode lengths of up to $6400$. Finally, we evaluate S5 on a challenging Meta-RL task in which the environment samples a random DMControl environment [40] and a random linear projection of the state and action spaces at the beginning of each episode. We show that the S5 agent achieves higher returns than LSTMs in this setting. Furthermore, we demonstrate that the resulting S5 agent performs well even on random linear projections of the state and action spaces of out-of-distribution held-out tasks.

## 2 Background

### 2.1 Structured State Space Sequence Models

State Space Models (SSMs) have been widely used to model various phenomenon using first-order differential equations [14]. At each timestep $t$, these models take an input signal $u(t)$. This is used to update a latent state $x(t)$ which in turn computes the signal $y(t)$. Some of the more widely-used

canonical SSMs are continuous-time linear SSMs, which are defined by the following equations:

$$\dot{x}(t) = \mathbf{A}x(t) + \mathbf{B}u(t)$$
$$y(t) = \mathbf{C}x(t) + \mathbf{D}u(t) \tag{1}$$

where $\mathbf{A}, \mathbf{B}, \mathbf{C}$, and $\mathbf{D}$ are matrices of appropriate sizes. To model sequences with a fixed step size $\Delta$, one can *discretise* the SSM using various techniques, such as the zero-order hold method, to obtain a simple linear recurrence:

$$x_n = \bar{\mathbf{A}}x_{n-1} + \bar{\mathbf{B}}u_n$$
$$y_n = \bar{\mathbf{C}}x_n + \bar{\mathbf{D}}u_n \tag{2}$$

where $\bar{\mathbf{A}}, \bar{\mathbf{B}}, \bar{\mathbf{C}}$, and $\bar{\mathbf{D}}$ can be calculated as functions of $\mathbf{A}, \mathbf{B}, \mathbf{C}, \mathbf{D}$, and $\Delta$.

S4 [12] proposed the use of SSMs for modelling long sequences and various techniques to improve its stability, performance, and training speeds when combined with deep learning. For example, S4 models use a special matrix initialisation to better preserve sequence history called HiPPO [11].

One of the primary strengths of the S4 model is that it can be converted to both a recurrent model, which allows for fast and memory-efficient inference-time computation, and a convolutional model, which allows for efficient training that is *parallelisable across timesteps* [13].

More recently, Smith et al. [38] proposed multiple simplifications to S4, called S5. One of its contributions is the use of *parallel scans* instead of convolution, which vastly simplifies S4's complexity and enables more flexible modifications. Parallel scans take advantage of the fact that the composition of *associative* operations can be computed in any order. Recall that for an operation $\bullet$ to be associative, it must satisfy $(x \bullet y) \bullet z = x \bullet (y \bullet z)$.

Given an associative binary operator $\bullet$ and a sequence of length $N$, parallel scan returns:

$$[e_1, e_1 \bullet e_2, \cdots, e_1 \bullet e_2 \bullet \cdots \bullet e_N] \tag{3}$$

For example, when $\bullet$ is addition, the parallel scan calculates the prefix-sum, which returns the running total of an input sequence. Parallel scans can be computed in $O(\log(N))$ time when given a sequence of length $N$, given $N$ parallel processors.

S5's parallel scan is applied to initial elements $e_{0:N}$ defined as:

$$e_k = (e_{k,a}, e_{k,b}) := (\bar{\mathbf{A}}, \bar{\mathbf{B}}u_k) \tag{4}$$

Where $\bar{\mathbf{A}}, \bar{\mathbf{B}}$, and $u_k$ are defined in Equation 2. S5's parallel operator is then defined as:

$$a_i \bullet a_j = (a_{j,a} \odot a_{i,a}, a_{j,a} \otimes a_{i,b} + a_{j,b}) \tag{5}$$

where $\odot$ is matrix-matrix multiplication and $\otimes$ is matrix-vector multiplication. The parallel scan then generates the recurrence in the hidden state $x_n$ defined in Equation 2.

$$e_1 = (\bar{\mathbf{A}}, \bar{\mathbf{B}}u_1) \qquad\qquad = (\bar{\mathbf{A}}, x_1) \tag{6}$$
$$e_1 \bullet e_2 = (\bar{\mathbf{A}}^2, \bar{\mathbf{A}}x_1 + \bar{\mathbf{B}}u_2) = (\bar{\mathbf{A}}^2, x_2) \tag{7}$$
$$e_1 \bullet e_2 \bullet e_3 = (\bar{\mathbf{A}}^2, \bar{\mathbf{A}}x_2 + \bar{\mathbf{B}}u_3) = (\bar{\mathbf{A}}^3, x_3) \tag{8}$$

Note that the model assumes a hidden state initialised to $x_0 = 0$ by initialising the scan with $e_0 = (\mathbf{I}, 0)$.

## 2.2 Reinforcement Learning

A Markov Decision Process (MDP) [39] is defined as a tuple $\langle \mathcal{S}, \mathcal{A}, R, P, \gamma \rangle$, which defines the environment. Here, $\mathcal{S}$ is the set of states, $\mathcal{A}$ the set of actions, $R$ the reward function that maps from a given state and action to a real value $\mathbb{R}$, $P$ defines the distribution of next-state transitions given a state and action, and $\gamma$ defines the discount factor. The agent's objective is to find a policy $\pi$ (a function which maps from a given state to a distribution over actions) which maximises the expected discounted sum of returns.

---

**Algorithm 1** Pseudocode for the Multi-Environment Meta-Learning environment step.

---

**Require:** Distribution of environments $E \sim \rho_E$, a fixed output observation dimension size $O$, and a fixed action dimension size $A$. Agent action $a$ and Environment termination $d$
1: **function** StepEnvironment($a$, $d$)
2:     **if** the environment terminated ($d$) **then**
3:         Sample random environment $E \sim \rho_E$
4:         Initialise random observation projection matrix $M_o \in \mathbb{R}^{E^o \times O}$ where $E^o$ is $E$'s observation size
5:         Initialise random action projection matrix
          $M_a \in \mathbb{R}^{A \times E^a}$ where $E^a$ is $E$'s action size
6:         Reset $E$ to receive an initial observation $o$
7:         Apply the random observation projection matrix to the observation $o' = M_o o$
8:         Append $r = 0$ and $d = 0$ to $o'$ to get $o''$
9:         Return $o''$
10:     **else**
11:         Apply the projection matrix $a' = M_a a$
12:         Step $E$ using $a'$ to receive the next observation $o$, reward $r$, and done signal $d$.
13:         Apply the projection matrix $o' = M_o o$
14:         Append $r$ and $d$ to $o'$ to get $o''$
15:         Return $o''$, $r$, and $d$
16:     **end if**
17: **end function**

---

$$\mathbb{E}[\mathbf{R}^\gamma | \pi] = \mathbb{E}_{s_0 \sim d, a_{0:\infty} \sim \pi, s_{1:\infty} \sim P} \Big[ \sum_{t=0}^\infty \gamma^t R(s_t, a_t) \Big]$$

In a Partially-Observed Markov Decision Process (POMDP), the agent receives an observation $o_t \sim O(s_t)$ instead of directly observing the state. Because of this, the optimal policy $\pi^*$ does not depend just on the current observation $o_t$, but (in generality) also on all previous observations $o_{0:t}$ and actions $a_{0:t}$.

## 3 Method

We first modify to S5 to handle variable-length sequences, which makes the architecture more suitable for tackling POMDPs. We then introduce a challenging new Meta-RL setting that tests for broad generalisation capabilities.

### 3.1 Resettable S5

Implementations of on-policy policy gradient algorithms with parallel environments often collect fixed-length trajectory "rollouts" from the environment for training, despite the fact that the environment episode lengths are often far longer and vary significantly. Thus, the collected rollouts (1) often begin within an episode and (2) may contain episode boundaries. Note that there are other, more complicated, approaches to rollout collection that can be used to collect full episode trajectories [24].

To handle trajectory rollouts that begin in the middle of an episode, sequence models must be able to access the state of memory that was present prior to the rollout's collection. Usually, this is done by storing the RNN's hidden state at the beginning of each rollout to perform truncated backpropagation through time [44]. This is challenging to do with transformers because they do not normally have an explicit recurrent hidden state, but instead simply retain the entire history during inference time. This is similarly challenging for S4 models since they assume that all hidden states are initialised identically to perform a more efficient backwards pass.

To handle episode boundaries within the trajectory rollouts, memory-based models must be able to reset their hidden state, otherwise they would be accessing memory and context from other episodes. RNNs can trivially reset their hidden state when performing backpropagation through time, and

transformers can mask out the invalid transitions. However, S4 models have no such mechanism to do this.

To resolve both of these issues, we modify S5's associative operator to include a reset flag that preserves the associative property, allowing S5 to efficiently train over sequences of varying lengths and hidden state initialisations. We create a new associative operator $\oplus$ that operates on elements $e_k$ defined:

$$e_k = (e_{k,a}, e_{k,b}, e_{k,c}) := (\bar{\mathbf{A}}, \bar{\mathbf{B}} u_k, d_k) \tag{9}$$

where $d_k \in \{0, 1\}$ is the binary "done" signal for the given transition from the environment.

We define $\oplus$ to be:

$$a_i \oplus a_j := \begin{cases} (a_{j,a} \odot a_{i,a}, a_{j,a} \otimes a_{i,b} + a_{j,b}, a_{i,c}) & \text{if } a_{j,c} = 0 \\ (a_{j,a}, a_{j,b}, a_{j,c}) & \text{if } a_{j,c} = 1 \end{cases}$$

We prove that this operator is associative in Appendix A. We now show that the operator computes the desired value. Assuming $e_{n,c} = 1$ corresponds to a "done" transition while all other timesteps before it ($e_{0:n-1,c} = 0$) and after it ($e_{n+1:L,c} = 0$) do not, we see:

$$\begin{aligned} e_0 \oplus \cdots \oplus e_{n-1} &= (\bar{\mathbf{A}}^{n-1}, \bar{\mathbf{A}} x_{n-2} + \bar{\mathbf{B}} u_{n-1}, 0) \\ &= (\bar{\mathbf{A}}^{n-1}, x_{n-1}, 0) \\ e_0 \oplus \cdots \oplus e_n &= (\bar{\mathbf{A}}, \bar{\mathbf{B}} u_n, 1) \\ &= (\bar{\mathbf{A}}, x_n, 1) \\ e_0 \oplus \cdots \oplus e_{n+1} &= (\bar{\mathbf{A}}^2, \bar{\mathbf{A}} x_n + \bar{\mathbf{B}} u_{n+1}, 1) \\ &= (\bar{\mathbf{A}}^2, x_{n+1}, 1) \end{aligned}$$

Note that even if there are multiple "resets" within the rollout, the desired value will still be computed.

### 3.2 Multi-Environment Meta-Learning with Random Projections

Most prior work in Meta-RL has only demonstrated the ability to *adapt* to a small range of similar tasks [2]. Furthermore, the action and observation spaces usually remain identical across different tasks, which severely limits the diversity of meta-learnning environments. To achieve more general meta-learning, ideally the agent should learn from tasks of varying complexity and dynamics. Inspired by Kirsch et al. [21], we propose taking *random linear projections* of the observation space and action space to a fixed size, allowing us to use the same model for tasks of varying observation and action space sizes. Furthermore, randomised projections *vastly* increase the number of tasks in the meta-learning space. We can then evaluate the ability of our model to *generalise* to unseen held-out tasks.

We provide pseudocode for the environment implementation in Algorithm 1.

## 4 Experiments

### 4.1 Memory Length Environment

First, we demonstrate our modified S5's improved training speeds in performance in the extremely simple memory length environment proposed in bsuite [32].

The environment is based on the well-known 't-maze' environment [31] in which the agent receives a cue on the first timestep, which corresponds to the action the agent should take some number of steps in the future to receive a reward. We run our experiments using bsuite's actor-critic baseline while swapping out the LSTM for Transformer self-attention blocks or S5 blocks [42] and using Gymnax for faster environment rollouts [22].

We show the results in Figure 1. In general, S5 displays a better asymptotic runtime than Transformers while far outperforming LSTMs in both performance and speed. Note that while S5 is theoretically

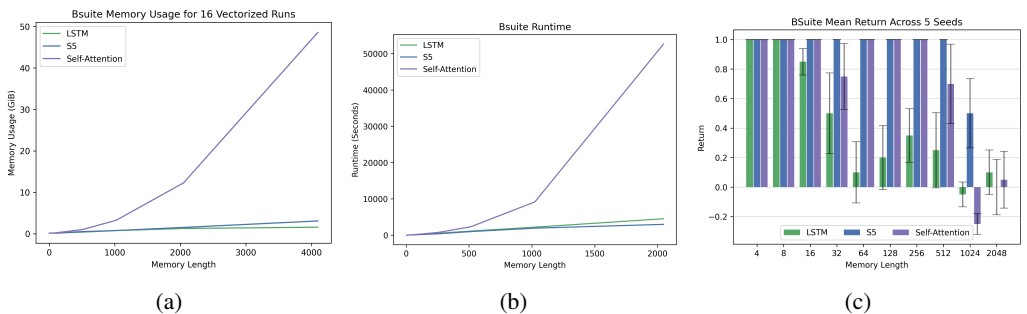

(a)                                         (b)                                         (c)

Figure 1: Evaluating S5, LSTM, and Self-Attention across different Bsuite memory lengths in terms of (a) memory usage, (b) runtime, and (c) return. Error bars report the standard error of the mean across 5 seeds. Runs were performed on a single NVIDIA A100.

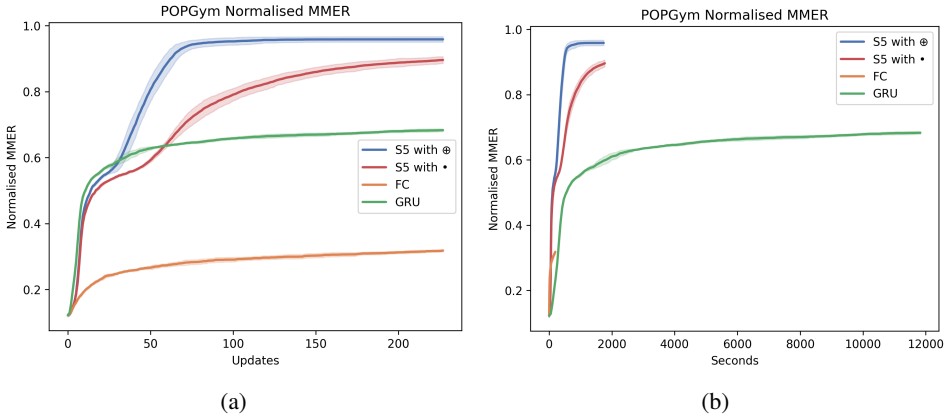

(a)                                                             (b)

Figure 2: (a) Results across implemented environments in POPGym's suite. Scores are normalised by the max-MMER per-environment. The shaded region represents the standard error of the mean across eight seeds. (b) The runtime for a single seed averaged across the environments for each architecture. Note that our implementation is end-to-end compiled to run entirely on a single NVIDIA A40.

$O(\log(N))$ in the backwards pass during training time (given enough processors), it is still bottle-necked by rollout collection from the environment, which takes $O(N)$ time. Because of the poor runtime performance of transformers for long sequences, we did not collect results for them in the following experiments.

## 4.2 POPGym Environments

We evaluate our S5 architecture on environments from the Partially Observable Process Gym (POP-Gym) [27] suite, a set of simple environments designed to benchmark memory in deep RL. To increase experiment throughput on a limited compute budget, we carefully re-implemented environments from the POPGym suite entirely in JAX [3] by leveraging existing implementations of CartPole and Pendulum in Gymnax [22]. PyTorch does not support the associative scan operation, so we could directly use the S5 architecture in POPGym's RLLib benchmark.

Morad et al. [27] evaluated several architectures and found the Gated Recurrent Unit (GRU) [5] to be the most performant. Thus, we compare our results to the GRU architecture proposed in the original POPGym benchmark. Note that POPGym's reported results use RLLib's [24] implementation of PPO, which makes several non-standard code-level implementation decisions. For example, it uses a dynamic KL-divergence coefficient on top of the clipped surrogate objective of PPO [37] – a feature that does not appear in most PPO implementations [9]. We instead use a recurrent PPO implementation that is more closely aligned with StableBaselines3 [35] and CleanRL's [18] recurrent PPO implementations. We include more discussion and the hyperparameters in Appendix B.

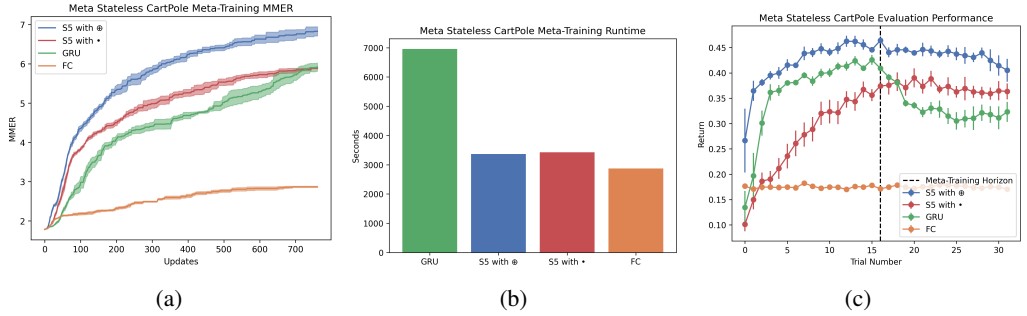

| (a) | (b) | (c) |

Figure 3: (a) Performance and (b) runtime on randomly-projected StatelessCartPole across 4 seeds. The shaded region represents the standard error. (c) Shows performance at the end of training across different trials. We evaluate on 32 trials even though we only train on 16. GRU's appear to have overfit to the training horizon while S5 models continue to perform well. The error bars represent the standard error across 4 seeds. Runs were performed on a single NVIDIA A100.

We show the results in Figure 2 and Appendix C. Notably, the S5 architecture performs well on the challenging "Repeat Previous Hard" task, far outperforming all architectures tested in Morad et al. [27]. Furthermore, the S5 architecture also runs over six times faster than the GRU architecture.

### 4.3 Randomly-Projected CartPole In-Context

We first demonstrate the long-horizon in-context learning capabilities of the modified S5 architecture by meta-learning across random projections of POPGym's Stateless CartPole (CartPole without velocity observations) task. More specifically, we perform the observation and action projections described in Section 3.2 and report the results in Figure 3.

Agents are given 16 trials per episode on randomly-sampled linear projections of StatelessCartPole's observation and action spaces. We find that S5 with $\oplus$ outperforms GRU's while running twice as quickly. Furthermore, we show that performance improves across trials, demonstrating in-context learning and show that the modified S5 architecture, unlike the GRU, can continue to learn even past its training horizon, which was previously shown using Transformers in Adaptive Agent Team et al. [1].

### 4.4 Multi-Environment Meta-Reinforcement Learning

We run our S5 architecture, an LSTM baseline, and a memory-less fully-connected network in the environment described in Section 3.2. For these experiments, we use Muesli [15] as our policy optimisation algorithm. We randomly project the environment observations to a fixed size of 12 and randomly project from an action space of size 2 to the corresponding environment's action space. We selected all of the DMControl environments and tasks that had observation and action spaces of size equal to or less than those values and split them into train and test set environments.

We use the S5 architecture described in Smith et al. [38], which consists of multiple stacked layers of S5 blocks. For both S5 and LSTM architectures, we found that setting the trajectory length equal to the maximum episode length $1k$ achieved the best performance in this setting. We provide a more detailed description of our

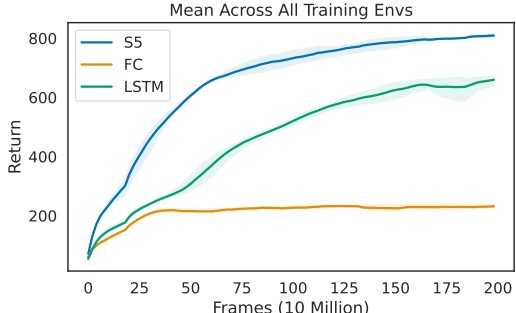

Figure 4: The mean of the return across all of the training environments. The shaded regions represent the range of returns reported across the three seeds. The environment observations and actions are randomly projected as described in Algorithm 1.

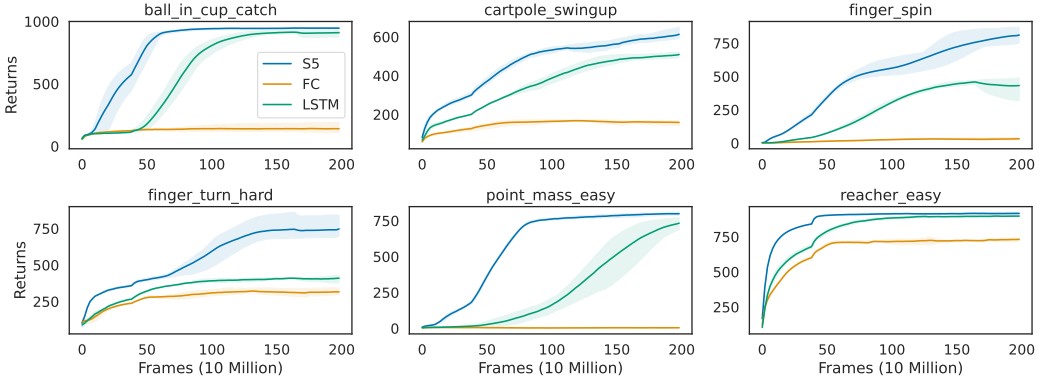

Figure 5: Results across the different environments of the training distribution. The shaded regions represent the range of returns reported across the three seeds. The environment observations and actions are randomly projected as described in Algorithm 1

architecture and hyperparameters in the supplementary material.

**In-Distribution Training Results** We meta-train the model across the six DMControl environments in Figure 5 and show the mean performance across them in Figure 4. Note that while these environments are in the training distribution, they are still being evaluated on *unseen random linear projections* of the state and action spaces and are *not given task labels*. Thus, the agent must infer from the reward and obfuscated dynamics which environment it is in. In this setting, S5 outperforms LSTMs in both sample efficiency and ultimate performance.

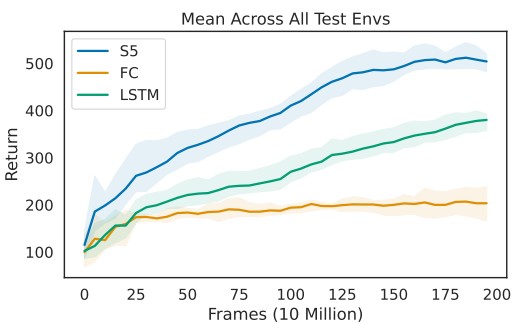

Figure 6: The mean of the return across all of the unseen held-out environments. The shaded regions represent the range of returns reported across the three seeds. The environment observations and actions are randomly projected as described in Algorithm 1.

**Out-of-Distribution Evaluation Results:** After training the model on the DMControl tasks in Figure 5, we next evaluate the trained model on random linear projections of five *held-out* DMControl tasks, *without any extra fine-tuning*. We present the mean score across these tasks in Figure 6 and the results for each task in Figure 7. While the agent displays impressive transfer performance to some tasks with unseen rewards and dynamics, it fails to successfully transfer to a completely unseen task in the test set, pendulum swingup, which has an unseen observation space, action space, and reward dynamics.

## 5 Related Work

S4 models have previously been shown to work in a number of previous settings, including audio generation [10] and video modeling [28]. Concurrent work investigated S4 models in other reinforcement learning settings. David et al. [6] investigated S4 models in a largely offline RL setting, with some modifications for online finetuning of the model. Morad et al. [27] investigated sequence models for POMDPs and found that naively using S4 models did not perform well.

Most works in memory-based meta-RL have not focused on generalisation to out-of-distribution tasks, but instead focused on maximising performance on the training task distribution [8, 43, 2]. Many past works that increased generalisation in meta-RL did so by restricting the model architecture class or architecture to symmetric models [20], loss functions [17], target values [30], or drift functions [25]. In contrast, this work achieves generalisation by vastly increasing the task distribution *without limiting the expressivity of the underlying model*, which eliminates the need for hand-crafted restrictions.

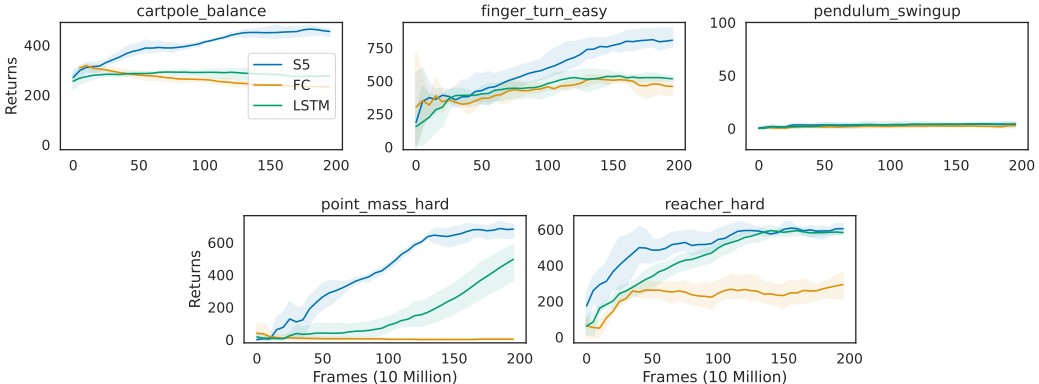

Figure 7: Results when evaluating on held-out DMControl tasks. The shaded regions represent the range of returns reported across the three seeds. The environment observations and actions are randomly projected as described in Algorithm 1

Some works perform long-horizon meta-reinforcement learning through the use of evolution strategies [17, 25]. This is because RNNs and Transformers have historically struggled to model very long sequences due to computational constraints and vanishing or exploding gradients [26]. However, evolution strategies are notoriously sample inefficient and computationally expensive.

Other works have investigated different sequence model architectures for memory-based reinforcement learning. Ni et al. [29] showed that using well-tuned RNNs can be particularly effective compared to many more complicated methods in POMDPs. Parisotto et al. [34] investigated the use of transformer-based models for memory-based reinforcement learning environments.

Sequence models have also been used for a number of other tasks in RL. For example, Transformers have been used for offline reinforcement learning [4], multi-task behavioural cloning [36], and algorithm distillation [23]. Concurrent work used transformers to also demonstrate out-of-distribution generalisation in meta-reinforcement learning by leveraging a large task space [1].

## 6   Conclusion and Limitations

In this paper, we investigated the performance of the recently-proposed S5 model architecture in reinforcement learning. S5 models are highly promising for reinforcement learning because of their strong performance in sequence modelling tasks and their fast and efficient runtime properties, with clear advantages over RNNs and Transformers. After identifying challenges in integrating S5 models into existing recurrent reinforcement learning implementations, we made a simple modification to the method that allowed us to reset the hidden state within a training sequence.

We then showed the desirable properties S5 models in the bsuite memory length task. We demonstrated that S5 is *asymptotically* faster than Transformers in the sequence length. Furthermore, we also showed that S5 models run nearly twice as quickly as LSTMs with the same number of parameters while outperforming them. We further evaluated our S5 architecture on environments in the POPGym suite [27], where we match or outperform RNNs while also running nearly five times faster. We achieve strong results in the "Repeat Previous Hard" task, which previous models struggled to solve.

Finally, we proposed a new meta-learning setting in which we meta-learn across random linear projections of the observation and action spaces of randomly sampled DMControl tasks. We show that S5 outperforms LSTMs in this setting. We then evaluate the models on held-out DMControl tasks and demonstrate out-of-distribution performance to unseen tasks through in-context adaptation.

There are several possible ways to further investigate S5 models for reinforcement learning in future work. For one, it may be possible to learn or distill [23] entire reinforcement learning algorithms within an S5 model, given its ability to scale to extremely long contexts. Another direction would be to investigate S5 models for continuous-time RL settings [7]. While $\Delta$, the discrete time between timesteps, is fixed for the original S4 model, S5 can in theory use a different $\Delta$ for each timestep.

**Limitations:** There are several notable limitations of this architecture and analysis. Firstly, implementing the associative scan operator is currently not possible using PyTorch, limiting us to using the JAX [3] framework. Furthermore, on tasks where short rollouts are sufficient to achieve good performance, S5 offers limited speedups, as rolling out across time is no longer a bottleneck. Finally, it was not possible to perform a fully comprehensive hyperparameter sweep in our results in Section 4.4 because the experiments used significant amounts of compute.

## Acknowledgments and Disclosure of Funding

Work funded by DeepMind. We would like to thank Antonio Orvieto, Robert Lange, Junhyok Oh, Greg Farquhar, Ted Moskovitz, and the rest of the Discovery Team at DeepMind for their helpful discussions throughout the course of the project.

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

# A Proof of Associativity of Binary Operator

Recall that $\oplus$ is defined as:

$$a_i \oplus a_j := \begin{cases} (a_{j,a} \odot a_{i,a}, a_{j,a} \otimes a_{i,b} + a_{j,b}, a_{i,c}) & \text{if } a_{j,c} = 0 \\ (a_{j,a}, a_{j,b}, a_{j,c}) & \text{if } a_{j,c} = 1 \end{cases}$$

This is equivalent to the following:

$$a_i \oplus a_j := \begin{cases} ((a_i \bullet a_j)_a, (a_i \bullet a_j)_b, a_{i,c}) & \text{if } a_{j,c} = 0 \\ a_j & \text{if } a_{j,c} = 1 \end{cases}$$

where $\bullet$ is S5's binary operator defined in Equation 5. Note that $\bullet$'s associativity was proved in Smith et al. [38]. Using this, we can prove the associativity of $\oplus$.

Let $x$, $y$, and $z$ refer to three elements. We will prove that for all possible values of $x$, $y$, and $z$, $\oplus$ retains associativity.

**Case 1:** $z_c = 1$

$$(x \oplus y) \oplus z = z \tag{10}$$
$$= y \oplus z \tag{11}$$
$$= x \oplus (y \oplus z) \tag{12}$$

**Case 2:** $z_c = 0$ and $y_c = 1$

$$(x \oplus y) \oplus z = y \oplus z \tag{13}$$
$$\text{Note that } (y \oplus z)_c = 1 \text{ thus,} \tag{14}$$
$$= x \oplus (y \oplus z) \tag{15}$$

**Case 3:** $z_c = 0$ and $y_c = 0$

$$(x \oplus y) \oplus z = ((x \bullet y)_a, (x \bullet y)_b, x_c) \oplus z \tag{16}$$
$$= (((x \bullet y) \bullet z)_a, ((x \bullet y) \bullet z)_b, x_c) \tag{17}$$
$$= ((x \bullet (y \bullet z))_a, (x \bullet (y \bullet z))_b, x_c) \tag{18}$$
$$= x \oplus ((y \bullet z)_a, (y \bullet z)_b, y_c) \tag{19}$$
$$= x \oplus (y \oplus z) \tag{20}$$

# B Hyperparameters

Table 2: Hyperparameters for training A2C on Bsuite

| Parameter | Value |
|---|---|
| Adam Learning Rate | 3e-4 |
| Entropy Coefficient | 0.0 |
| Encoder Layer Sizes | [256, 256] |
| Number of Recurrent Layers | 1 |
| Size of Recurrent Layer | 256 |
| Discount $\gamma$ | 0.99 |
| TD $\lambda$ | 0.9 |
| Number of Environments | 1 |
| Unroll Length | 32 |
| Number of Episodes | 10000 |
| Activation Function | ReLU |

Table 3: Hyperparameters for training PPO on POPGym

| Parameter | Value |
|---|---|
| Adam Learning Rate | 5e-5 |
| Number of Environments | 64 |
| Unroll Length | 1024 |
| Number of Timesteps | 15e6 |
| Number of Epochs | 30 |
| Number of Minibatches | 8 |
| Discount $\gamma$ | 0.99 |
| GAE $\lambda$ | 1.0 |
| Clipping Coefficient $\epsilon$ | 0.2 |
| Entropy Coefficient | 0.0 |
| Value Function Weight | 1.0 |
| Maximum Gradient Norm | 0.5 |
| Learning Rate Annealing | None |
| Activation Function | LeakyReLU |
| Encoder Layer Sizes | [128, 256] |
| Recurrent Layer Hidden Size | 256 |
| Action Decoder Layer Sizes | [128, 128] |
| Value Decoder Layer Sizes | [128, 128] |
| S5 Layers | 4 |
| S5 Hidden Size | 256 |
| S5 Discretization | ZOH |
| S5 $\Delta$ min | 0.001 |
| S5 $\Delta$ max | 0.1 |

Table 4: Hyperparameters for training Muesli on Multi-Environment Meta-RL. These experiments were run using 64 TPUv3's.

| Parameter | Value |
|---|---|
| Adam Learning Rate | 3e-4 |
| Value Function Weight | 1.0 |
| Muesli Auxiliary Loss Weight | 3.0 |
| Muesli Model Unroll Length | 1.0 |
| Encoder Layer Sizes | [512, 512] |
| Number of Environments | 1024 |
| Discount $\gamma$ | 0.995 |
| Rollout Length | 1000 |
| Offline Data Fraction | 0.0 |
| Total Frames | 2e9 |
| LSTM Hidden Size | 512 |
| Projected Observation Size | 12 |
| Projected Action Size | 2 |
| S5 Layers | 10 |
| S5 Hidden Size | 256 |
| S5 Discretization | ZOH |
| S5 $\Delta$ min | 0.001 |
| S5 $\Delta$ max | 0.1 |

## C  POPGym Discussion

|  | Stateless CartPole Hard | Noisy Stateless CartPole Hard | Stateless Pendulum Hard | Noisy Stateless Pendulum Hard | Repeat Previous Hard |
|---|---|---|---|---|---|
| S5 (ours) | $\mathbf{1.0 \pm 0.0}$ | $\mathbf{0.28 \pm 0.0}$ | $\mathbf{0.79 \pm 0.01}$ | $0.55 \pm 0.01$ | $\mathbf{0.91 \pm 0.01}$ |
| GRU (ours) | $\mathbf{1.0 \pm 0.0}$ | $0.27 \pm 0.0$ | $0.75 \pm 0.0$ | $\mathbf{0.61 \pm 0.01}$ | $-0.46 \pm 0.01$ |
| MLP (ours) | $0.26 \pm 0.0$ | $0.22 \pm 0.0$ | $0.41 \pm 0.02$ | $0.34 \pm 0.01$ | $-0.48 \pm 0.00$ |
| GRU | $\mathbf{1.000 \pm 0.000}$ | $0.390 \pm 0.007$ | $\mathbf{0.828 \pm 0.001}$ | $\mathbf{0.657 \pm 0.002}$ | $-0.428 \pm 0.002$ |
| MLP | $0.265 \pm 0.002$ | $0.229 \pm 0.002$ | $0.477 \pm 0.030$ | $0.351 \pm 0.012$ | $-0.486 \pm 0.002$ |
| IndRNN | $\mathbf{1.000 \pm 0.000}$ | $\mathbf{0.404 \pm 0.005}$ | $0.804 \pm 0.023$ | $0.521 \pm 0.109$ | $-0.384 \pm 0.013$ |
| LMU | $0.987 \pm 0.007$ | $0.352 \pm 0.019$ | $0.806 \pm 0.006$ | $0.563 \pm 0.014$ | $\mathbf{0.191 \pm 0.041}$ |
| S4D | $0.127 \pm 0.026$ | $0.207 \pm 0.007$ | $0.303 \pm 0.014$ | $0.289 \pm 0.011$ | $-0.491 \pm 0.001$ |
| FART | $\mathbf{0.996 \pm 0.000}$ | $0.366 \pm 0.002$ | $0.698 \pm 0.077$ | $0.553 \pm 0.007$ | $-0.485 \pm 0.001$ |

Table 5: Results in POPGym's suite. The reported number is the max-mean episodic reward (MMER) used in Morad et al. [27]. To calculate this, we take the mean episodic reward for each epoch, and then take the maximum over all epochs. For our results above, the mean and standard deviation across eight seeds are reported. The results below are selected architectures from Morad et al. [27], which also includes the best-performing one from each environment. They report the mean and standard deviation across three trials.

Morad et al. [27] used RLLib's [24] implementation of PPO, which differs significantly from standard implementations of PPO. It uses a dynamic KL-divergence coefficient on top of the clipped surrogate objective of PPO [37]. Furthermore, they use advanced orchestration to return full episode trajectories, rather than using the more commonly-studied "stored state" [19] strategy.

Instead, we follow the design decisions outlined in the StableBaselines3 [35] and CleanRL's [18] recurrent PPO implementations. While this results in different results shown in Table 5 for the same architecture, it recovers similar performance across the environments. Notably, our S5 architecture far outperforms the best performing architecture in Morad et al. [27] in the "RepeatPreviousHard" environment.

We used the learning rate, number of environments, unroll length, timesteps, epochs, and minibatches, GAE $\lambda$, and model architectures from Morad et al. [27]. However, we used the standard clipping coefficient $\epsilon$ of $0.2$ instead of $0.3$ to account for the lack of a dynamic KL-divergence coefficient. Note that we also adjusted the S5 architecture to contain approximately the same number of *parameters* as the GRU implementation instead of matching the size of the *hidden state*, which was done in Morad et al. [27].

We did not evaluate our architecture across the full POPGym suite. To enable more rapid experimentation, we implement our algorithms and environments end-to-end in JAX [3, 25]. While the original POPGym results took 2 hours per trial with a GRU with a Quadro RTX 8000 and 24 CPUs, we could run our experiments using only 3 *minutes* per trial on an NVIDIA A40. Because of this, we selected environments from Morad et al. [27] to implement in JAX. We chose the CartPole, Pendulum, and Repeat environments because they are modified versions of existing environments in Lange [22]. We found that the "Easy" and "Medium" versions of these environments were not informative, as most models perform well on them and only report the "Hard" difficulty results.

We attach our code in the supplementary materials.

