# OpenReview forum: "Structured State Space Models for In-Context Reinforcement Learning"
_NeurIPS.cc/2023/Conference — NeurIPS 2023 poster_

### Official Review · Reviewer_gUv9 · 2023-07-03

**Soundness:** 3 good
**Presentation:** 3 good
**Contribution:** 3 good
**Rating:** 7
**Confidence:** 3

**Summary:**

The authors propose a modification to S5 that enables "resetting" the recurrent state, allowing it to function as an RNN replacement in RL. They update the scan operator to utilize the `done` flag and use this to reset the recurrent state. They evaluate the resettable S5 on a portion of the POPGym suite and the bsuite memory task. They show that the S5 is able to solve long-term memory tasks previous methods were unable to solve.

Next, they implement a metalearning approach from Kirsch et al. that enables learning across various action and observation space sizes by using random linear projections of the spaces to a fixed-size vector. They leverage S5's long-term memory to obtain good results across DMLab, on both in-distribution and out-of-distribution tasks.

**Strengths:**

The paper is well written and addresses a painful issue: that partially observable policies are inefficient to train. They show that S5 is the first model to solve the difficult RepeatHard task from POPGym. It is clear the author put a ton of work into implementation and put care into baseline implementations to ensure a fair comparison. The experimental setup is quite broad in the sense that it covers both POMDPs and metalearning.

**Weaknesses:**

#### Main weakness:
Perhaps I am missing some information, and if so, please correct me. But the reset, the main contribution of the paper, appears to be a trivial change to S5. Line 94 already provides the initial state of vanilla S5, which is $e_0 = (I, 0)$. Their reset is just setting the recurrent state to the initial state when they receive a `done` flag from the environment. This is the standard for any recurrent model in RL, so I do not believe this counts as a novel contribution.

With a slight change of notation, the S5 scan operator is

$f(a_t, b_t, a_{t-1}, b_{t-1}) = \begin{bmatrix}
a_t \odot a_{t-1} &
a_t \otimes b_{t-1} + b_t
\end{bmatrix}$

The authors propose that at $t=0$, since we do not have $a_{t-1}, b_{t-1}$ we evaluate $f$ using $a_{t-1} = I$ and $b_{t-1} = 0$

$f(a_t, b_t, I, 0) = \begin{bmatrix}
a_t &
b_t
\end{bmatrix}$

So to reset: just put the predefined vanilla S5 initial state $e_0$ and into S5.

#### Other Weaknesses
- The authors show very promising performance on Repeat Hard and a few stateless cartpole environments. That said, POPGym has something like 50 total tasks, so it is a bit disappointing that they picked the easiest control tasks, as I imagine S5 would do much better than GRUs on harder tasks.

**Questions:**

- The reuse of $a$ is confusing in the scan terms, since $a$ is also the subscript referring to the $A$ matrix. Perhaps a different term would be more clear?
- Figure 3: This is across all POPGym envs or just a subset? From briefly reviewing the code I assume a subset.
- I'm not an expert on meta rl, but it is very promising that their approach can do well on out-of-distribution tasks.
- Experiments and implementation are the strength of this paper, I just wish they just put the S5 reset in the appendix or footnote

**Limitations:**

The authors do not have a limitations section. It is probably worth adding a short section on this.

---

> ### Author Rebuttal · Authors · 2023-08-08
>
> We would like to first thank the reviewer for their detailed and technical review. We are glad that the reviewer finds our approach promising and the experiments fair and broad.
>
> ### On the Reset
>
> >the reset, the main contribution of the paper, appears to be a trivial change to S5
>
> It’s not immediately obvious why the parallel reset is non-trivial. Indeed, for any recurrent model in RL, resetting the hidden state during inference (and training) is standard. However, it’s not clear how to do this when one is *parallelizing across the time dimension* during *training/backpropagation* given a batch of data of shape $(Minibatch Size, Sequence Length, Observation Size)$. For example, most S4-like models usually implement their backwards pass using *convolutions* (instead of recurrence) to parallelize across time -- it’s not clear how or if we can adjust the convolution kernel to account for “dones” from the environment.
>
> Instead, S5 uses *associative scans*, which leverages a *binary associative operator* to parallelize across the time dimension. While it’s immediately obvious that simple operators, such as multiplication and addition, can be implemented associatively, previous implementations have not shown how to implement “reset” function associatively. Our contribution is introducing such an operator (Line 132) that computes the desired output (Line 135) and is provably associative (Appendix A).
>
> With this in mind:
>
> >So to reset: just put the predefined vanilla S5 initial state $e_0$ into S5
>
> We assume the reviewer is asking why we cannot just insert $e_0$ where there are “dones”. We believe this would not compute the desired recurrence. While indeed (using the reviewer’s notation) $f(a_t, b_t, I, 0)$ computes the desired value, note that $f(I, 0, a_{t-1}, b_{t-1})$ (i.e. the other side of the associative operation) returns $[a_{t-1} b_{t-1}]$. Thus, when scanning over the sequence $(e_{t-1}, e_0, e_t)$, we would get $f(a_t, b_t, a_{t-1}, b_{t-1}) \neq [a_t, b_t]$. This means it does not reset the hidden state. At no point would inserting $e_0$ actually *reset* the hidden state of the scan -- it would merely repeat the previous element.
>
> Furthermore, inserting elements into the scan would usually involve dynamic shapes or masking, which is often inefficient or undesirable when performing batch learning.
>
> We have updated our manuscript to clarify why we cannot just use $e_0$ to perform the reset
>
> ---
>
> *Because this was the reviewer’s primary concern, we hope that our response has clarified this contribution and that the reviewer will let us know of any further concerns on this topic or, otherwise, consider updating their score.*
>
> ---
>
> > The reuse of $a$ is confusing in the scan terms
>
> Good catch! We’ve updated the paper to reflect this change
>
> ### On the tasks
>
> > POPGym has something like 50 total tasks, so it is a bit disappointing that they picked the easiest control tasks
> > This is across all POPGym envs or just a subset?
>
> Indeed, this is just a subset of the POPGym envs. While POPGym does have many tasks, many of them are simply easier versions of the ones we evaluated on. For example, we evaluate only on the “Hard” version of these tasks while there are “Medium” and “Easy” versions that are not particularly informative. Unfortunately, there are no harder control tasks in POPGym beyond CartPole and Pendulum; however, we evaluated on the environment that most architectures struggled the most on: RepeatPreviousHard. The best-performing architecture in POPGym reported a score of 0.191 while we obtained a score of 0.91.
>
> We’ve since added further additional results on a long-horizon version of POPGym StatelessCartPole that involves sequences of length up to $6400$ to increase the difficulty and context. We’ve included a brief explanation and analysis in the general response.
>
> > It is probably worth adding a short section on [limitations].
>
> Good point -- thanks for the feedback! We have included this in our manuscript.

---

> > ### Comment · Reviewer_gUv9 · 2023-08-11
> >
> > Thanks, I did not consider the other side of the scan.
> >
> > The Bsuite, long-term CartPole, and RepeatHard task performance is impressive, but I was hoping to see how S5 performs on a broader range of POMDPs. Looking at the POPGym website, they appear to have tasks like Minesweeper or Autoencode which I would argue test different memory capabilities than inferring position from velocity or store/recall from RepeatHard.
> >
> > Your results are good, I just think they could be more convincing if your model was demonstrated on more tasks. I have updated my score accordingly.

---

> > > ### Author Response · Authors · 2023-08-15
> > > **Additional POPGym Results**
> > >
> > > We thank the reviewer for the fast response, and for updating their score. We’re happy that the reviewer finds the performance on the Bsuite, long-term CartPole, and RepeatHard to be impressive, though we understand that the reviewer is disappointed that we did not evaluate on the full POPGym suite.
> > >
> > > To further address the reviewer’s concerns, we implemented as many of the POPGym environments as we could in pure JAX. We believe that this in and of itself is a significant contribution, and will provide and open-source the code upon acceptance (thanks to the reviewer). This will speed up research in partially-observable RL significantly, since it allows researchers to run statistically-significant experiments in minutes rather than several hours.
> > >
> > > On top of the “Repeat”, “Cartpole,” and “Pendulum” environments in the original paper, we’ve now also implemented the “Minesweeper”, “Higher Lower”, “Count Recall”, “Autoencode”, “Multiarmed Bandit”, and “Concentration” environments.
> > >
> > > This only means we have not yet implemented 2 environments: “Battleship” and “Labyrinth”, which would take more effort. We plan to implement these before the camera-ready version. Should the reviewer believe that these environments are crucial to their evaluation (or should we finish implementing early), we will report the results before the Reviewer Discussion deadline.
> > >
> > > We only report on the “Hard” difficulty of the *new* environments for brevity here. Results for "Medium" and "Easy" are similar and we can report them should the reviewer request them. We ran 4 vectorized seeds on an NVIDIA A100 to get the following results:
> > >
> > > ## MMER
> > > | Method            | Minesweeper        | Higher Lower      | Count Recall      | Autoencode        | Multiarmed Bandit | Concentration     |
> > > |-------------------|--------------------|-------------------|-------------------|-------------------|-------------------|-------------------|
> > > | S5 with $\oplus$  | **-0.296 ± 0.002** | **0.505 ± 0.001** | **-0.833 ± 0.000** | **-0.296 ± 0.002** | **0.562 ± 0.019** | **-0.831 ± 0.001**|
> > > | S5 with $\bullet$| -0.345 ± 0.003     | 0.499 ± 0.000     | -0.877 ± 0.000    | -0.345 ± 0.003     | 0.438 ± 0.019     |**-0.831 ± 0.001**  |
> > > | GRU              | -0.313 ± 0.003     | **0.505 ± 0.000** | **-0.832 ± 0.001**| -0.313 ± 0.003     | **0.575 ± 0.008** | **-0.830 ± 0.001**|
> > > | MLP              | -0.383 ± 0.004     | **0.504 ± 0.000** | -0.877 ± 0.000    | -0.383 ± 0.004     | 0.306 ± 0.012     | -0.832 ± 0.000    |
> > >
> > >
> > > ## Runtime (Seconds) for 4 Vectorized Seeds
> > >
> > > | Method            | Minesweeper   | Higher Lower | Count Recall | Autoencode    | Multiarmed Bandit | Concentration |
> > > |-------------------|---------------|--------------|--------------|--------------|-------------------|---------------|
> > > | S5 with $\oplus$  | 1030.565      | 1016.776     | 1038.935     | 1031.494     | 1458.849          | 1043.843     |
> > > | S5 with $\bullet$| 939.419       | 927.000      | 953.165      | 940.351      | 1359.034          | 954.630      |
> > > | GRU              | 10309.002     | 10379.492    | 10716.769    | 10585.172    | 10452.576         | 9775.959     |
> > > | MLP              | 172.651       | 157.777      | 179.184      | 173.515      | 586.735           | 233.702      |
> > >
> > >
> > > Notably, because we are running on an A100 instead of an A40, we achieve significantly faster speeds for S5, nearly 10x that of the GRU in these environments while obtaining very similar results across the board.
> > >
> > > We do not expect the S5 architecture to necessarily outperform the GRU significantly since these environments do not test long-term memory; however, it is notable that S5 still runs approximately 10x faster while achieving similar results.
> > >
> > > We hope that these latest results address the reviewer’s latest concerns. We would be happy to engage in further discussion with the reviewer or perform more experiments. We would like to once again thank the reviewer for the timely and thoughtful feedback, which we believe has effectively strengthened the paper's quality and contributions.

---

> > > > ### Comment · Reviewer_gUv9 · 2023-08-15
> > > >
> > > > Thank you for addressing our concerns. Reimplementing POPGym in Jax is quite a bit of work and an arguably significant contribution if jittable and open-sourced. I will update the score once more in good faith, given the author's progress. I ask that:
> > > >
> > > > 1. The authors add scores for all additional environments to the paper.
> > > > 2. The environment code is publicly released in a usable state to the community. That is, there are some unit tests and there is enough documentation for others to get started with their code.

---

> > > > > ### Author Response · Authors · 2023-08-15
> > > > > **Thank you for the fast response!**
> > > > >
> > > > > We would like to once again greatly thank the reviewer for their extremely fast reply and updated score. We commit to adding scores for all additional environments to the paper, and open-sourcing the code in a usable state before the camera-ready deadline if accepted.
> > > > >
> > > > > We are glad the reviewer agrees that reimplementing POPGym in JAX is an arguably significant contribution. We believe that the fast S5 architecture will massively speedup research in RL tasks that are partially-observable while the fast JAX-jittable POPGym implementation is a neat addition that will speed up research in understanding and evaluating new algorithms and architectures.

---

### Official Review · Reviewer_R3ow · 2023-07-04

**Soundness:** 2 fair
**Presentation:** 2 fair
**Contribution:** 2 fair
**Rating:** 5
**Confidence:** 3

**Summary:**

This paper investigates the effectiveness of structured state-space sequence (S4) models and in particular its variant S5 in reinforcement learning settings. To apply S5 to reinforcement learning, the authors propose a modified associative operator that handles episodic resets, allowing S5 to train over sequences spanning multiple episodes. Across a suite of partially observed RL environments from bsuite and POPGym, S5 outperforms RNN and Transformer baselines while being significantly more efficient at training and inference. To further evaluate the S5's ability to perform in-context RL and generalize out-of-distribution, the authors introduce a multi-environment meta-learning task based on DM Control where each meta-episode features different random projections of observation and action spaces. Not only does S5 outperform MLP and LSTM on in-distribution tasks, but it demonstrates better generalization to held-out tasks as well.

**Strengths:**

- The paper proposes a novel associative operator which allows the S5 model to handle episodic resets in RL settings. In doing so, they demonstrate the effectiveness of the S5 model for partially-observed and meta-RL tasks, both in terms of asymptotic performance and training/inference speed.
- The meta-RL environment with random observation and action projections can be used as an evaluation benchmark for future work.

**Weaknesses:**

- The proposed method is a rather marginal modification to the S5 model. And according to the results in Table 2, the new operator does not significantly outperform the vanilla S5 operator.
- It would be fine to thoroughly evaluate an existing method in a new setting, making it an empirical analysis paper. However, the experiments in this paper are insufficient. The bsuite memory length is a toy environment. And there is a lack of baselines and variety of tasks in the POPGym experiments. It is elusive why the authors include MLP but not other baselines from [1].
- A major benefit of state-space models is their ability to handle significantly longer sequences than RNNs and Transformers, but the paper does not demonstrate an RL setting where this can be useful.
- The meta-learning setting, while novel, feels a bit contrived. It is unclear what the implications of this setting are.

[1] Steven Morad, Ryan Kortvelesy, Matteo Bettini, Stephan Liwicki, and Amanda Prorok. POPGym: Benchmarking partially observable reinforcement learning. In The Eleventh International Conference on Learning Representations, 2023.

**Questions:**

- What is the reason behind the improved OOD generalization of the S5 model compared to baselines?
- From Table 2, the proposed operator does not significantly outperform the vanilla S5 operator. Can you include more comparisons with the vanilla operator? E.g. by evaluating the vanilla S5 model on the meta-learning tasks.
- According to [1], S4D is the worst-performing method out of all baselines. Is there a reason behind the huge improvement in asymptotic performance?

[1] Steven Morad, Ryan Kortvelesy, Matteo Bettini, Stephan Liwicki, and Amanda Prorok. POP- Gym: Benchmarking partially observable reinforcement learning. In The Eleventh International Conference on Learning Representations, 2023.




**Limitations:**

The authors have not adequately addressed the limitations in the paper. I would imagine the limitations to be similar to those of state-space models. For example, they generally perform worse than RNN on partially observed RL settings, and they are hard to scale to higher dimensions.

---

> ### Author Rebuttal · Authors · 2023-08-08
>
> We thank the reviewer for their insightful feedback. We are happy to hear that the reviewer agrees that we “demonstrate the effectiveness of the S5 model for partially-observed and meta-RL tasks, both in terms of asymptotic performance and training/inference speed” and that “the meta-RL environment with random observation and action projections can be used as an evaluation benchmark for future work.” We would like to address the reviewer’s concerns.
>
> > The proposed method is a rather marginal modification to the S5 model...in Table 2, the new operator does not significantly outperform the vanilla S5 operator
> > From Table 2, the proposed operator does not significantly outperform the vanilla S5 operator.
>
> In Table 2, we show that the new operator significantly outperforms the vanilla S5 operator in $3/5$ of the environments, and matches in the remaining $2/5$. In the most challenging environment, the new operator raises the score from $0.76$ to $0.91$. We believe this is very significant, and visualize it more clearer in Figure 10 of our rebuttal.
>
> We show this further in the shared rebuttal. The Meta-StatelessCartPole (Figure 8a and 8b) experiments further demonstrate its significance on a task that requires longer memory.
>
> Furthermore, state-space models in on-policy RL in general have not been rigorously studied and are not widely-used. The operator is just one of the contributions. We hope this paper and these results can encourage further adoption of this architecture to speedup and improve future research involving memory-based architectures for RL.
>
> > It is elusive why the authors include MLP but not other baselines from [1].
>
> We report GRU results since it was generally the most consistent best-performing baseline evaluated in [1]. It is also one of the most widely-used. We include the MLP to demonstrate the performance vs. runtime tradeoff in Figure 3.
>
> > A major benefit of state-space models is their ability to handle significantly longer sequences than RNNs and Transformers, but the paper does not demonstrate an RL setting where this can be useful.
>
> Indeed, that is *a* benefit of state-space models. There are *many other benefits* of state-space models, many of which we show in the paper! In particular, we get *significant speedups* (over 6x faster on POPGym!) while also *significantly outperforming the baselines*.
>
> ***We have also attached further additional results to show a scenario where longer sequence adaptation of up to $6400$ is useful. Please read the general rebuttal for plots (in Figure 8) and experimental setup***.
>
> > The meta-learning setting, while novel, feels a bit contrived. It is unclear what the implications of this setting are.
>
> Most existing RL settings do not require long sequence lengths (arguably *because* existing architectures cannot handle them), so we designed this meta-learning setting to construct one that does.
>
> In the future, a more scaled up version of this setting with many more tasks and data augmentation could result in a *general* meta-learned reinforcement learning agent that can adapt to much farther OOD tasks.
>
> > What is the reason behind the improved OOD generalization of the S5 model compared to baselines?
>
> We believe it is because the S5 model can incorporate information over longer sequences, which we demonstrated in the Bsuite experiments (Figure 2). The ability to incorporate longer contexts could have led to a more general adaptation mechanism than one that can only adapt to recent transitions.
>
> > According to [1], S4D is the worst-performing method out of all baselines. Is there a reason behind the huge improvement in asymptotic performance?
>
> We hypothesize that it is because their implementation only uses one layer of S4D. While for most recurrent networks, stacking additional recurrent layers does not help in many tasks, including RL (see the architectures used for [2] and [3]), for state-space models it is extremely important for performance since it is a completely linear recurrence (and can only achieve non-linearity through depth).
>
> >limitations to be similar to those of state-space models. For example, they generally perform worse than RNN on partially observed RL settings, and they are hard to scale to higher dimensions.
>
> We agree with the reviewer and have since added a limitation section, which includes implementation challenges (e.g. the lack of associative scans in PyTorch) and potentially diminishing speedup returns on tasks with short sequences or other computational bottlenecks (such as vision-based tasks). However, we are not sure why the reviewer believes that SSM’s perform worse than RNN’s on partially observed RL settings or that they are hard to scale to higher dimensions. Our experiments show that they perform significantly better on partially observed RL settings. Other concurrent work [5] has shown similar results in Offline RL and have scaled them to higher dimensions [4].
>
> ---
> *We hope that most of the reviewer’s concerns have been addressed and, if so, they would reconsider their assessment. We’d be happy to engage in further discussions.*
>
> ---
>
> [1] Steven Morad, Ryan Kortvelesy, Matteo Bettini, Stephan Liwicki, and Amanda Prorok. POP- Gym: Benchmarking partially observable reinforcement learning. In The Eleventh International Conference on Learning Representations, 2023.
>
> [2] OpenAI, C. Berner, et al. "Dota 2 with large scale deep reinforcement learning." arXiv preprint arXiv:1912.06680 2 (2019).
>
> [3] Vinyals, Oriol, et al. "Grandmaster level in StarCraft II using multi-agent reinforcement learning." Nature 575.7782 (2019): 350-354.
>
> [4] Deng, Fei, Junyeong Park, and Sungjin Ahn. "Facing off World Model Backbones: RNNs, Transformers, and S4." arXiv preprint arXiv:2307.02064 (2023).
>
> [5] David, Shmuel Bar, et al. "Decision S4: Efficient Sequence-Based RL via State Spaces Layers." The Eleventh International Conference on Learning Representations. 2022.

---

> > ### Comment · Reviewer_R3ow · 2023-08-15
> >
> > Thank you for the additional experiments and clarifications. From Fig. 3 of the rebuttal it does seems that the proposed associative operator leads to an overall improvement over vanilla S5, which addresses my main concern. While I still view this paper as more of an empirical evaluation of S5 in RL settings, I believe this is a reasonable contribution and will adjust my score.

---

> > > ### Author Response · Authors · 2023-08-15
> > > **Thank you for the response! Have you looked at our new POPGym results?**
> > >
> > > We would like to greatly thank the reviewer for acknowledging our rebuttal and adjusting their score.
> > >
> > > We do not wish to take up more of the reviewer’s time, **but we would like to confirm if the reviewer has also had a to chance to read our [latest response to Reviewer gUv9](https://openreview.net/forum?id=4W9FVg1j6I&noteId=13iOA4ADUV), which includes new results on many more POPGym environments.** We copy the key section below.
> > >
> > > ---
> > > ## Additional POPGym Environments
> > >
> > > To further address the reviewer’s concerns, we implemented as many of the POPGym environments as we could in pure JAX. We believe that this in and of itself is a significant contribution, and will provide and open-source the code upon acceptance (thanks to the reviewer). This will speed up research in partially-observable RL significantly, since it allows researchers to run statistically-significant experiments in minutes rather than several hours.
> > >
> > > On top of the “Repeat”, “Cartpole,” and “Pendulum” environments in the original paper, we’ve now also implemented the “Minesweeper”, “Higher Lower”, “Count Recall”, “Autoencode”, “Multiarmed Bandit”, and “Concentration” environments.
> > >
> > > This only means we have not yet implemented 2 environments, “Battleship” and “Labyrinth”, which would take more effort. We plan to implement these before the camera-ready version. Should the reviewer believe that these environments are crucial to their evaluation (or should we finish implementing early), we will report the results before the Reviewer Discussion deadline.
> > >
> > > We only report on the “Hard” difficulty of the *new* environments for brevity here. Results for "Medium" and "Easy" are similar and we can report them should the reviewer request them. We ran 4 vectorized seeds on an NVIDIA A100 to get the following results:
> > >
> > >
> > > ### MMER
> > > | Method            | Minesweeper        | Higher Lower      | Count Recall      | Autoencode        | Multiarmed Bandit | Concentration     |
> > > |-------------------|--------------------|-------------------|-------------------|-------------------|-------------------|-------------------|
> > > | S5 with $\oplus$  | **-0.296 ± 0.002** | **0.505 ± 0.001** | **-0.833 ± 0.000** | **-0.296 ± 0.002** | **0.562 ± 0.019** | **-0.831 ± 0.001**|
> > > | S5 with $\bullet$| -0.345 ± 0.003     | 0.499 ± 0.000     | -0.877 ± 0.000    | -0.345 ± 0.003     | 0.438 ± 0.019     |**-0.831 ± 0.001**  |
> > > | GRU              | -0.313 ± 0.003     | **0.505 ± 0.000** | **-0.832 ± 0.001**| -0.313 ± 0.003     | **0.575 ± 0.008** | **-0.830 ± 0.001**|
> > > | MLP              | -0.383 ± 0.004     | **0.504 ± 0.000** | -0.877 ± 0.000    | -0.383 ± 0.004     | 0.306 ± 0.012     | -0.832 ± 0.000    |
> > >
> > >
> > > ### Runtime (Seconds) for 4 Vectorized Seeds
> > > | Method            | Minesweeper   | Higher Lower | Count Recall | Autoencode    | Multiarmed Bandit | Concentration |
> > > |-------------------|---------------|--------------|--------------|--------------|-------------------|---------------|
> > > | S5 with $\oplus$  | 1030.565      | 1016.776     | 1038.935     | 1031.494     | 1458.849          | 1043.843     |
> > > | S5 with $\bullet$| 939.419       | 927.000      | 953.165      | 940.351      | 1359.034          | 954.630      |
> > > | GRU              | 10309.002     | 10379.492    | 10716.769    | 10585.172    | 10452.576         | 9775.959     |
> > > | MLP              | 172.651       | 157.777      | 179.184      | 173.515      | 586.735           | 233.702      |
> > >
> > >
> > > Notably, because we are running on an A100 instead of an A40, we achieve significantly faster speeds for S5, nearly 10x that of the GRU in these environments while obtaining very similar results across the board.
> > >
> > > We do not expect the S5 architecture to necessarily outperform the GRU significantly since these environments do not test long-term memory; however, it is notable that S5 still runs approximately 10x faster while achieving similar results.
> > >
> > > ---
> > >
> > > > I still view this paper as more of an empirical evaluation of S5 in RL settings
> > >
> > > This is a very fair interpretation! Hopefully our additional results on Meta-StatelessCartPole and our six additional POPGym environments would further the strength of the empirical evaluations, which we know was also one of the reviewer's key concerns.
> > >
> > > We hope that these results further address the reviewer’s remaining concerns. We would be happy to engage in further discussion with the reviewer and would like to thank the reviewer for their feedback, which we believe has strengthened the paper significantly.

---

### Official Review · Reviewer_Zjtv · 2023-07-05

**Soundness:** 2 fair
**Presentation:** 1 poor
**Contribution:** 3 good
**Rating:** 5
**Confidence:** 3

**Summary:**

Structured state space sequence (S4) models deliver good performance on long-range sequence modeling tasks, fast inference speed and parallelize training, making them suitable for many RL settings. The authors propose a modification to the recently proposed S5 architecture and apply it to RL tasks. Their proposed model outperforms Transformers in terms of runtime and memory complexity (on a toy task), and RNNs in terms of task performance. They aim to show the efficacy of SSMs for in-context adaptation to new task variations.

**Strengths:**

- The authors are the first (to the best of our knowledge) to leverage S4-like models for RL. This is the main novelty of the paper, and we consider this an important contribution.
- The authors propose a modification/fix for the S5 architecture to enable handling sequences of varying lengths in and RL setting.
- They show that S4-based models have advantages over commonly used Transformers (runtime, memory complexity) and RNNs (task performance) on a toy task (Figure 2).


**Weaknesses:**

**Major concerns**:

While the method to add a mechanism for resetting the state of S5 layers seems sound the proposed evaluation and in-context RL setting do not make sense to me.
The authors themselves state, that using S5 layers for RL is problematic as such models would be "accessing memory and context from other episodes".
Why then evaluate the proposed solution in the in-context setting, where accessing context (e.g. from other episodes) is required?

Furthermore, I do not understand what the authors mean with "in-context RL", and apart from the conclusion the authors never even mention it again even though its in the title.
Instead, the authors perform meta-RL experiments by training models on multiple tasks simultaneously, however even here I don't understand the evaluation protocol.
I would not call fine-tuning for another 2 billion timesteps an "evaluation".
How does this experiment show "in-context adaptation"?

Finally, the authors argue that comparison to transformer based models is not possible due to poor runtime performance.
However, at least for the POPGym environment this should have been possible, seeing as the authors state in the appendix that the runtime per trial with GRU is only 3 minutes.
And for in-context experiments a transformer-based baseline should definitely be included.

**Minor comments:**

- Algorithm 1 seems redundant and can be moved to the Appendix, as it only shows a meta-RL loop with random projections.
- Figure 1 is too large for its information content.
- Line 166: “mmemory”



**Questions:**

- Figure 2:
  - How do models perform beyond a sequence length of 512? S5 and self-attention reach the same level of performance at all lengths. How does this change for longer sequence lengths (at least up to 2048)?
    Do Transformers outperform S5 beyond some context length?
    No standard deviations, only single seed?
  - How does the memory consumption differ between the compared methods?

- Figure 3, Table 2:
  - What sequence length is used? Is a long sequence length even required for these tasks?
  - The performance gains of S5 over GRU come from the “Repeat Previous Hard” task. Why is this the case? On all other tasks, they perform the same. Seems like the benchmark is too easy to solve and not a suitable test-bed.
  - Why are the standard deviations 0.0 for all methods on cartpole? Seems unlikely across 8 seeds (especially in the noisy setting).

- Section 4.3:
  - The setup for this experiment needs clarification:
    - Why does this experiment demonstrate in-context learning abilities? Does the model always only observe observations from the current episode?
    - Are you training on held-out tasks for 2B steps (as shown on the x-axis) or just doing inference? From the learning curves, it looks like weights are updated. Why is this “evaluation” then? Please clarify.
    - How would an agent trained from scratch compare against the pre-trained one? Learning the new tasks within 2B steps should not be an issue.
    - Are you resetting S5 at episode boundaries during evaluation? Wouldn’t it be beneficial to maintain the context across episodes to encourage in-context learning abilities? It seems like resetting is a disadvantage here.
    - How would the agents perform if trained and evaluated without random projections?

  - Figure 7:
    - In general, the considered tasks are very similar to the pre-training tasks. Four of the five tasks are minor variations of them. Is it correct to consider them OOD?
    - Importantly, the model fails completely on pendulum_swingup. This suggests that, the task distribution is too narrow. Conducting this experiment on a larger number of tasks or robot morphologies may be more insightful.
    - Why is “finget_turn_easy” in the OOD tasks, but the hard variant “finget_turn_hard” in the pre-training tasks. Shouldn't it be the other way round?
  - Missing ablation on the proposed modification to S5. How does performance change when using the proposed modification vs. without?
  - Please report parameter counts of the compared methods for all experiments (only reported for POPGym in the Appendix). Do all architectures have approximately the same amount of parameters?


**Limitations:**

- Limitations of the proposed architecture have not been discussed sufficiently. As there is no comparison against the Transformer architecture, it is hard to assess the limitations of the proposed architecture.

---

> ### Author Rebuttal · Authors · 2023-08-08
>
> We would like to first thank the reviewer for their extremely thorough review. We are glad that the reviewer finds that investigating S4-like models for RL is an important contribution, especially since they have not been widely-adopted or thoroughly investigated in RL.
>
> ### In-Context Learning and Meta-Learning
>
> > Why evaluate...where accessing context (e.g. from other episodes) is required?
> > Are you resetting S5 at episode boundaries during evaluation?
>
> RL^2-like meta-RL methods, an “episode” can consist of multiple “trials” (the terminology on this is inconsistent between papers in the field). In meta-RL it is important to access context between different *trials* but not *episodes*. We have updated the manuscript to clarify this.
>
> In our meta-RL setting we only allow the agent access to **one trial** because it already achieves near-optimal performance in the underlying tasks through within-trial adaptation. ***We have attached additional results to show multi-trial in-context adaptation in the general rebuttal  Figure 8.***
>
> > I do not understand what the authors mean with "in-context RL"...How does this experiment show "in-context adaptation"?
>
> While in-context learning is now commonly used to refer to few-shot learning in transformers, we are referring to *in-context changes in behavior (adaptation)* when referring to “in-context RL”. We hope Figure 8c makes this clearer, where we show the agent learns across trials!
>
> > Are you training on held-out tasks for 2B steps?
> > How would an agent trained from scratch compare?
>
> Those plots are for evaluation throughout *meta-training*. The held-out tasks are thus evaluated at different points throughout *meta-training*. During the evaluation, we are just doing inference (including on held-out tasks). An agent trained from scratch could not solve an out-of-distribution task in one episode.
>
> > The considered tasks are very similar to the pre-training tasks
>
> It would require a very large collection of pre-training tasks (or inductive bias on the meta-learner) to enable any transfer to far out-of-distribution tasks. We believe that showing transfer to close, but still OOD tasks, is still a positive result that shows robustness and generalization.
>
> > Why is “finget_turn_easy” in the OOD tasks?
>
> The “easy” and “hard” versions of tasks are variants of the task with different sets of parameters or dynamics. Transferring either way demonstrates OOD performance. The choice of which versions were “held-out” was arbitrarily selected.
>
> > How does performance change when using the proposed modification vs. without?
>
> The experiments for the original DMControl suite are slow and expensive to run. We have instead performed a similar experiment using random linear projections of POPGym’s StatelessCartPole, which we share in the general response Fig 8. The results show that performance improves significantly when using the proposed modification.
>
> > comparison to transformer based models
>
> Transformer-based models are far too slow for us to run in the meta-DMControl setting and are an *uncommon architecture choice for on-policy RL* because of their runtime. While transformers can show fast training on supervised learning tasks, they are slow in reinforcement learning due to their poor inference speed, especially when using 2B frames.
>
> > same amount of parameters?
>
> One can reconstruct the sizes from the hyperparameters.
>
> S5 Encoder: 1975040 parameters
>
> LSTM Encoder: 2099200 parameters
>
> ### BSuite Results
>
> > How do models perform beyond...(at least up to 2048)?...only single seed?
>
> The original bsuite memory length evaluation only goes up to $105$. $512$ is already nearly 5x that number. Credit assignment gets increasingly infeasible as the length gets longer, especially since bsuite evaluation uses a fixed number ($10000$) of episodes for training, a fixed $\gamma=0.99$, and a basic A2C algorithm. We are unlikely to get *any meaningful information* about an architecture’s capabilities past 512 steps.
>
> Furthermore, we expect self-attention to grow *cubically* in runtime since the number of steps grows *linearly*, and the cost-per-step grows *quadratically*. Sequence length $4096$ would be expected to take several days for self-attention. We are currently running these experiments.
>
> In the meantime, we have reported results for sequences up to $2048$ (Fig 9). Our original results reported the median score across $5$ seeds. Our updated ones in Figure 9c report the mean and standard error across $5$ seeds.
>
> > How does the memory consumption differ?
>
> We report this in the rebuttal! Figure 9a.
>
> ### POPGym Results
>
> > What sequence length is used? Is a long sequence length even required?
>
> A sequence length of 1024 is used. We follow the training hyperparameters outlined in POPGym [1]. Even if long sequence lengths are not required, S5 *still trains six times faster and outperforms the baselines*. POPGym is not designed to test learning across long sequences; however, it is useful for evaluating architectures for partially-observed RL. We have experiments in Figure 8 that require long sequences.
>
> > The performance gains of S5 over GRU come from the “Repeat Previous Hard” task. Why is this the case?
>
> We’re not sure why S5 performs better. POPGym’s best performing architecture, the LMU (MMER of $0.19$ vs. S5’s $0.91$), shares the underlying theory of continuous time representations.
>
> > Why are the standard deviations 0.0...on cartpole?
>
> The choice of hyperparameters leads to very consistent policy updates since they train for many epochs on large batch sizes. For StatelessCartPole, all methods achieve perfect score. For NoisyStatelessCartPole, there are very small standard deviations at more digits.
>
> [1] Steven Morad, et al. “POP- Gym: Benchmarking partially observable reinforcement learning.”
>
> ---
> *We hope that most of the reviewer’s concerns have been addressed and, if so, they would reconsider their assessment. We’d be happy to engage in further discussions.*

---

> > ### Comment · Reviewer_Zjtv · 2023-08-16
> >
> > We thank the authors for their effort during the rebuttal. We appreciate the clarifications and additional experiments. Therefore, we decided to update our score accordingly.

---

> > > ### Author Response · Authors · 2023-08-16
> > >
> > > We would like to thank the reviewer for reading our rebuttal and acknowledging our improvements, clarifications, and additional experiments. We would like to also further thank the reviewer for updating their score accordingly.
> > >
> > > We would be more than happy to discuss any remaining concerns or reservations the reviewer may have about the submission that is preventing them from increasing the score further.

---

### Official Review · Reviewer_R54Y · 2023-07-05

**Soundness:** 3 good
**Presentation:** 3 good
**Contribution:** 3 good
**Rating:** 7
**Confidence:** 4

**Summary:**

The authors propose a modification of the S5 sequence architecture with a resettable hidden state that leads to a drop-in replacement of RNNs and Transformers in partially observed / memory-intensive RL tasks. They show that S5 exceeds baseline performance while being more computationally efficient to train than LSTMs due to S5’s parallelization and faster at inference than Transformers’ N^2 attention.

**Strengths:**

State space models are a natural fit for in-context meta-RL and long-term memory because of their unique combination of train-time parallelism and test-time constant inference. They are also thought to be more compatible with the continuous inputs seen in RL than other long-sequence domains like NLP that use discrete tokens. In-context RL has a sequence length barrier that goes somewhat unnoticed because common benchmarks do not require adaptation over more than a few steps. Applying S5 and future versions here may have great long-term benefits.

The experiments do clearly demonstrate that resettable S5 can replace LSTMs and Transformers in partially observed tasks.

The paper also proposes a meta-learning version of the DM Control suite using randomly projected state and action spaces. While this is a somewhat arbitrary way to expand the meta-distribution of tasks from a small set of popular benchmarks, it is probably better than the common alternative of turning similar environments into goal-conditioned tasks and then hiding the goal (Ant Goal, Cheetah Fwd-Back, Humanoid Dir, ...). This benchmark will be useful for future work.

The method could be applied to most agents that support S5's computation, and should be easy to use and reproduce.

**Weaknesses:**

S5 would be best suited for extending the sequence lengths used by current methods, and could unlock a new level of difficulty in long-context RL. However, the experiments here stop short of exploring those limits. Instead, sequence lengths are capped at ~1k where LSTMs/Transformers are feasible but slower and more expensive. Wall-clock efficiency is a nice advantage but could also be improved by other factors like using a more efficient base algorithm. If the goal was to show that S5 is a valid substitute at existing sequence lengths while leaving expansion for future work, it would have been simpler to use more standard benchmarks in the meta-RL portion and compare to external baselines.

In general, the narrative of the paper gets a bit lost in the gray area between more traditional long-term memory and “in-context” meta-learning. The need for long sequences is motivated by multi-episodic in-context learners like RL^2 (lines 36-42), but the method (Sec. 3.1 and Alg 1) does not distinguish between episode resets (which would not reset the sequence model’s hidden state) and task resets. The first two experiments are focused on standard long-term memory, while the third appears to only evaluate zero-shot generalization to a meta-distribution of tasks. It would have been more interesting to evaluate RL^2-style multi-episodic learning, which would naturally extend the sequence length to highlight S5’s core advantages.

The need for a resettable hidden state within S5 is presented as a widespread barrier to its application in RL. But this is primarily caused by the data collection and optimization implementations common to on-policy policy gradient algorithms with parallel actors. I think it would be helpful to explain this issue in more detail than is provided (lines 111-144, 43-48). For example, there would be no need for the modified architecture when substituting S5 for RNNs/Transformers in an off-policy in-context learner like in [Ni et al.](https://arxiv.org/abs/2110.05038) The issue may also be avoided by batching the policy updates into padded sequences that do not cross episode boundaries.

**Questions:**

Please clarify if I am wrong in thinking that the DMControl tasks are zero-shot. The evaluation procedure is not as clearly discussed as it often is in meta-RL papers, and the appendix did not give more details.

Were there any experiments to dramatically increase the context length towards the range seen with S4 in LRA? I’m curious if there are insights on limitations or changes that would be needed to make this work in an RL setting. According to Table 5 the DM Control experiments used 64 TPUv3s, so I assume compute was not the limiting factor here.

**Limitations:**

There is no Limitations section or a clear discussion on limitations. However, the paper is proposing an architectural change to an existing method that has known limitations, and some of those limitations are improved by the S5 model.

---

> ### Author Rebuttal · Authors · 2023-08-08
>
> We would like to thank the reviewer for their well-thought review. The reviewer brings up many good points that we would like to address.
>
> Firstly, we are glad that the reviewer finds that our paper clearly demonstrates that resettable S5 can replace LSTM’s and Transformers and that our proposed benchmark can be used for future works. We would like to also highlight that our code is provided in the supplementary material, which should further help with reproducibility and ease of use.
>
> ## On Longer Contexts
>
> > sequence lengths are capped at ~1k where LSTMs/Transformers are feasible but slower and more expensive
>
> > It would have been more interesting to evaluate RL^2-style multi-episodic learning, which would naturally extend the sequence length to highlight S5’s core advantages.
>
> These are fair points! We originally wished to find benchmarks where longer sequence lengths would be helpful; however, we struggled to find scalable benchmarks (i.e. beyond POPGym), likely because current architectures would not be able to take advantage of the increased context. Because of this, we introduced our new meta-learning version of the DMControl Suite. We had expected that more in-context adaptation trials would be required because of the large space of tasks; however, our architecture was still able to perform near-optimally with just one trial. *We have since developed and evaluated the architecture on a version of StatelessCartPole where we simply perform random linear projections of the observation.* This involves sequences of length of up to $6400$ and we analyze its performance across multiple adaptation trials. We encourage the reviewer to read our general rebuttal and attached plots in Figure 8.
>
> >Were there any experiments to dramatically increase the context length towards the range seen with S4 in LRA? I’m curious if there are insights on limitations or changes that would be needed to make this work in an RL setting.
>
> We agree with the reviewer that long-range tasks, especially meta-RL tasks, would be ideal for investigating dramatically increased context lengths. Firstly, we struggled to find appropriate benchmarks that require many thousands of steps of context. Ni et al. [1] find that for the vast majority of common POMDPs (including meta-RL tasks), short context lengths are often optimal -- and anything over just $100$ is considered “long”. It’s unclear if this is because long contexts are generally not useful, or if current benchmarks were designed with existing architectures in mind. Secondly, in such settings, credit assignment and long-horizon discounting become increasingly challenging. For example, the common discount factor of 0.99 (and even 0.999) vanishes after several thousands of timesteps. We have since added a task that uses sequence length of up to $6400$ in Figure 8 of our general rebuttal.
>
> ## Others
>
> > But this is primarily caused by the data collection and optimization implementations common to on-policy policy gradient algorithms with parallel actors. I think it would be helpful to explain this issue in more detail than is provided
>
> This is a good point. The explanation in the current paper can be unclear (and it is impressive that the reviewer understands the nuances of these implementation details given that we did not fully explain them). We have since re-written this section to further explain why on-policy algorithms with parallel actors often make use of resettable hidden states and thank the reviewer for the suggestion. Indeed, in the off-policy setting, implementations have more control over data organization.
>
> > The issue may also be avoided by batching the policy updates into padded sequences that do not cross episode boundaries.
>
> Yes, this is also a potential approach -- however, padded sequences often involve large amounts of inefficient computation and would be significantly slower. If one is running on-policy algorithms they are usually concerned with wallclock-time rather than just sample efficiency. We’ve updated the paper to discuss this!
>
> > the DMControl tasks are zero-shot. The evaluation procedure is not as clearly discussed as it often is in meta-RL papers, and the appendix did not give more details.
>
> Yes, the DMControl tasks are zero-shot, with no fine-tuning involved. The agent is given only a single trial to adapt in-context. We have updated the manuscript to clarify this point further. Thank you for the suggestion!
>
> [1] Ni, Tianwei, Benjamin Eysenbach, and Ruslan Salakhutdinov. "Recurrent model-free rl can be a strong baseline for many pomdps." arXiv preprint arXiv:2110.05038 (2021).
>
> ---
> *We hope that most of the reviewer’s concerns have been addressed and, if so, they would consider updating their score. We’d be happy to engage in further discussions.*

---

> > ### Comment · Reviewer_R54Y · 2023-08-18
> >
> > Thanks for the detailed rebuttal and new results on a tight deadline. The long sequence length results look good. I don't think stateless cartpole and bsuite genuinely require adaptation over these context lengths, but the runtime advantages of S5 are clear. The lack of long-horizon domains is outside the scope of this paper, and if anything this work contributes to the idea that meta-RL has a serious benchmarking problem at the moment. I'd encourage the authors to prioritize open-source usability of the core model for future work. If S5 can be a stepping stone to enable longer sequences and shorter runtimes in meta-RL, then it should be as easy as possible to plug resettable S5 into other RL frameworks and experiments.
> >
> > The writing changes you discuss seem helpful. In general, the importance of the *resettable* aspect of S5 is a bit overstated by the original draft, given that this is a problem created by other implementation details that could be avoided. However this is still a nice feature that makes S5 compatible with popular high-throughput RL libraries.
> >
> > Could you clarify the comparison between the two S5 operators in results like Table 2 and the new Figure 8? The other operator is worse because it is not reset between task boundaries, so is trained on sequences that run through consecutive tasks? It might be clearer to rename it from the operator symbol to the concept ("S5 Without Resets" or similar).
> >
> > I will update my score to accept.

---

> > > ### Author Response · Authors · 2023-08-19
> > >
> > > We would like to thank the reviewer for their time, response, and update. Their feedback throughout this process has been extremely accurate and insightful. Many of the points the reviewer brought up aligned closely with our own thoughts about the initial draft and were articulated *very* clearly. The reviewer further brought up many good points around benchmarks in meta-RL and potentially unclear sections of the writing that we had not noticed.
> > >
> > > > don't think stateless cartpole and bsuite genuinely require adaptation over these context lengths
> > >
> > > We think bsuite and stateless cartpole test different times of long-horizon behavior, but *we generally agree with the reviewer that neither are fully ideal*. In particular, bsuite tests recall (which is not long-horizon adaptation) and the random projection stateless cartpole tests the ability to integrate information across a long-horizon (but does not require specific recall). Ideally a good environment would test both simultaneously.
> > >
> > > > the idea that meta-RL has a serious benchmarking problem at the moment
> > >
> > > We fully agree with the reviewer! As our architectures, algorithms, and computers get more powerful, the benchmarks also need to advance to keep up.
> > >
> > > > I'd encourage the authors to prioritize open-source usability of the core model for future work
> > >
> > > We will heavily prioritize this. We think our implementation (if the reviewer had a chance to look at the provided code) is generally accessible and *highly performant*, hopefully enabling wide scale experimentation and adoption.
> > >
> > > > the importance of the resettable aspect of S5 is a bit overstated by the original draft, given that this is a problem created by other implementation details that could be avoided
> > >
> > > This is fair! We think given the extent of our new results, we can likely also shorten this section in the paper for space and thus relegate its importance.
> > >
> > > > The other operator is worse because it is not reset between task boundaries, so is trained on sequences that run through consecutive tasks? It might be clearer to rename it
> > >
> > > Yes the reviewer’s understanding is correct -- we will rename it for the camera-ready copy! The reviewer brings up a good point -- this would be a much more clear naming since in theory we could have used the original operator but unrolled it in sequence like an RNN in order to perform the resets.
> > >
> > > We would like to once again thank the reviewer for their feedback!

---

### Author Rebuttal · Authors · 2023-08-08

We are grateful to the reviewers for their insightful feedback. We appreciate the consensus that the proposed S5 architecture offers **clear advantages over standard RNNs and Transformers in partially-observed RL, both in terms of performance and runtime.** This is the key takeaway of our work, and we hope it will accelerate future research in RL.

$\color{red} R1$ (R54Y): “experiments do clearly demonstrate that resettable S5 can replace LSTMs and Transformers in partially observed tasks”

$\color{green} R2$ (Zjtv): “They show that S4-based models have advantages over commonly used Transformers (runtime, memory complexity) and RNNs (task performance) on a toy task.”

$\color{blue} R3$ (R3ow): “they demonstrate the effectiveness of the S5 model for partially-observed and meta-RL tasks, both in terms of asymptotic performance and training/inference speed.”

$\color{magenta} R4$ (gUv9): “S5 is able to solve long-term memory tasks previous methods were unable to solve.”

We are glad that reviewers found the associative operator to be a “novel” ($\color{blue} R3$) modification that enables “drop-in replacement of RNN’s” ($\color{red} R1$) by “handling sequences of varying lengths” ($\color{green} R2$), although there are concerns with its significance ($\color{blue} R3$, $\color{magenta} R4$). We added Figure 10 to the general rebuttal to demonstrate its impact, along with results in Figure 8.

Some reviewers also found the meta-learning setup to be useful for future work ($\color{red} R1$, $\color{blue} R3$) and the results “very promising” ($\color{magenta} R4$), although some understandably have concerns over its clarity ($\color{green} R2$) and limited single-trial sequence length ($\color{red} R1$, $\color{green} R2$, $\color{blue} R3$) of $1000$, which we address below.

The results on POPGym demonstrated over *6x speedups* and better performance (being the “first model to solve the difficult RepeatHard task from POPGym” [$\color{magenta} R4$]) over the state-of-the-art GRU. Reviewers had concerns over the lack of Transformer-based baselines ($\color{green} R2$), lack of difficult tasks ($\color{green} R2$, $\color{blue} R3$, $\color{magenta} R4$), and short context lengths ($\color{red} R1$), which we address below.

## Meta StatelessCartPole

To address concerns about sequence length, in-context learning, and POPGym tasks, *we have run additional experiments combining POPGym’s StatelessCartPole task with randomly-projected observations.* We allow the agent to have $16$ trials per episode. We show in-context learning results in Figure 8(c) and demonstrate that it can even maintain performance to $32$ trials, evaluating on sequence lengths of up to *6400* -- which is a very long sequence length for reinforcement learning. We show training results and runtime in Figure 8(a) and 8(b) -- S5 still outperforms GRU’s while running significantly faster.

We hope this addresses the reviewer’s concerns around the lack of multiple trials, long context lengths, and challenging tasks.

## Transformers

To further address $\color{green} R2$’s concerns about the lack of transformer-based baselines, we included longer context results in bsuite Figure 9 for lengths up to $2048$ and include memory usage statistics. Note that bsuite’s original evaluation only extends to a length of $105$, and is likely unsuitable for such long sequences (since it uses a fixed episode budget). We are currently running even longer sequences; however, they are expected to take days to complete and will not change the takeaway.

Furthermore, we ran a Transformer baseline on POPGym environments that have *fixed episode lengths*. We then modify the trajectory collection such that it *always collects exactly one episode per rollout* while maintaining an identical total batch size. Without this constraint, it is challenging to efficiently run Transformers in on-policy RL while maintaining its memory. We perform the same for other architectures across three seeds and report the mean and standard error below. Transformers are fast in these settings since the horizons are short and they can parallelize across time (unlike LSTM’s).

| Architecture | stateless pendulum hard MMER | noisy stateless pendulum hard MMER | repeat previous hard MMER |
|---|---|---|---|
|S5|**0.805±0.003**|0.575±0.002|**0.986±0.000**|
|Transformer|**0.796±0.005**|0.544±0.003|-0.462±0.003|
|GRU|**0.808±0.005**|**0.628±0.001**|-0.457±0.013|

| Architecture | stateless pendulum hard (s) | noisy stateless pendulum hard (s) | repeat previous hard (s) |
|---|---|---|---|
|S5|**297.56±0.45**|**321.18±0.15**|**308.50±0.31**|
|Transformer|2018.25±0.22|706.49±0.10|863.64±0.20|
|GRU|970.82±11.60|1925.91±32.59|1476.80±16.37|


## Misc

We’ve also made numerous other changes and additions to our manuscript from the reviewer’s suggestions, including:

- Clarifying “the data collection and optimization implementations common to on-policy policy gradient algorithms with parallel actors” ($\color{red} R1$)

- The training and evaluation procedure of the DMControl multi-task setup ($\color{red} R1$, $\color{green} R2$)

- What we mean by “in-context learning” ($\color{green} R2$)

- Various fixes to spelling, notation, and resizing of figures ($\color{green} R2$, $\color{magenta} R4$)

- An explicit and thorough limitations section ($\color{red} R1$,$\color{green} R2$,$\color{blue} R3$,$\color{magenta} R4$)

- Clarifying why standard naive resetting does not work ($\color{magenta} R4$)

- Parameter counts for different architectures ($\color{green} R2$)

We hope these modifications adequately address the concerns raised by the reviewers, and we are confident they strengthen our manuscript's overall quality. To reiterate: We show that SSM’s provide *clear advantages in partially-observed RL* and this can accelerate future work in RL.

---

> ### Author Response · Authors · 2023-08-15
> **Sharing Additional Results on POPGym and BSuite (from Individual Responses).**
>
> We would like to thank the reviewers for their time, consideration, and feedback, which has strengthened our manuscript.
>
> $\color{magenta} R4$ believes “the task performance is impressive” and that our recent JAX-based POPGym implementation is an “arguably significant contribution if jittable and open-sourced”, **updating their score by two points** to *accept*.
>
> $\color{blue} R3$ agrees that “the proposed associative operator leads to an overall improvement over vanilla S5”, **updating their score by one point** to *borderline accept*. We hope $\color{blue} R3$ will also have a chance to take a look at our additional empirical results here.
>
> $\color{green} R2$ “appreciate[s] the clarifications and additional experiments”, **updating their score by two points** to *borderline accept* without raising any further or remaining concerns.
>
> $\color{red} R1$ agrees that "the long sequence length results look good" and that the writing changes "seem helpful", **updating their score by one point** to *accept*.
>
> We thank the reviewers for being so responsive and receptive -- the discussions have been incredibly useful. We summarize additional results here.
>
> ## Additional POPGym environments.
>
> In response to $\color{magenta} R4$’s and $\color{blue} R3$’s points that we only ran on a small set of POPGym, we have since implemented and evaluated *six more POPGym environments in pure JAX*. We’ve now also implemented the “Minesweeper”, “Higher Lower”, “Count Recall”, “Autoencode”, “Multiarmed Bandit”, and “Concentration” environments.
>
> This only means we have not yet implemented 2 environments: “Battleship” and “Labyrinth”, which would take more effort.
>
> We report on the “Hard” difficulty of the *new* environments for brevity here. We ran 4 vectorized seeds on an NVIDIA A100 to get the following results:
>
> ### MMER
> | Method| Minesweeper| Higher Lower| Count Recall| Autoencode| Multiarmed Bandit| Concentration|
> |---|---|---|---|---|---|---|
> | S5 with $\oplus$| **-0.296 ± 0.002** | **0.505 ± 0.001** | **-0.833 ± 0.000** | **-0.296 ± 0.002** | **0.562 ± 0.019** | **-0.831 ± 0.001**|
> | S5 with $\bullet$| -0.345 ± 0.003| 0.499 ± 0.000| -0.877 ± 0.000| -0.345 ± 0.003| 0.438 ± 0.019|**-0.831 ± 0.001**  |
> | GRU| -0.313 ± 0.003| **0.505 ± 0.000** | **-0.832 ± 0.001**| -0.313 ± 0.003| **0.575 ± 0.008** | **-0.830 ± 0.001**|
> | MLP| -0.383 ± 0.004| **0.504 ± 0.000** | -0.877 ± 0.000| -0.383 ± 0.004| 0.306 ± 0.012| -0.832 ± 0.000|
>
>
> ### Runtime (Seconds) for 4 Vectorized Seeds
>
> | Method| Minesweeper| Higher Lower| Count Recall| Autoencode| Multiarmed Bandit| Concentration|
> |---|---|---|---|---|---|---|
> | S5 with $\oplus$| 1030.565| 1016.776| 1038.935| 1031.494| 1458.849| 1043.843|
> | S5 with $\bullet$| 939.419| 927.000| 953.165| 940.351| 1359.034| 954.630|
> | GRU| 10309.002| 10379.492| 10716.769| 10585.172| 10452.576| 9775.959|
> | MLP| 172.651| 157.777| 179.184| 173.515| 586.735| 233.702|
>
>
> Notably, because we are running on an A100 instead of an A40, we achieve significantly faster speeds for S5, nearly 10x that of the GRU in these environments while obtaining very similar results across the board. We do not expect the S5 architecture to necessarily outperform the GRU significantly since these environments do not test long-term memory; however, it is notable that S5 still runs approximately 10x faster while achieving similar results.
>
> ## Additional Bsuite results
>
> Our Transformer-based run of Bsuite memory chain of length $4096$ has completed, which we ran to further address $\color{green} R2$’s concerns. We would like to emphasize our point that the Bsuite results and evaluation setup is unsuitable for such long horizons, since it trains on only $10,000$ episodes with a fixed $\gamma$ and basic A2C algorithm. Nonetheless, we find the runtimes to be informative.
>
> The latest Transformer run was $56$ times slower than the equivalent S5 run, taking nearly $4$ days to complete as opposed to $1.6$ hours on an NVIDIA A100.
>
> ### Runtime (Seconds) for 5 Vectorized Seeds
> |Architecture|4|8|16|32|64|128|256|512|1024|2048|4096|
> |---|-|-|--|--|--|---|---|---|----|----|----|
> |S5|12.9s|21.3s|32.3s|67.3s|104.4s|191.8s|383.6s|984.7s|1935.6s|2981.3s|5914.2s|
> |Transformer|14.7s|23.5s|43.7s|87.5s|174.0s|337.3s|765.3s|2293.7s|9165.7s|52649.3s|335979.5s|
> |LSTM|14.4s|22.2s|39.1s|72.6s|138.4s|270.4s|539.5s|1118.6s|2209.4s|4528.5s|7912.8s|
>
> ### Mean Return and Standard Error for 5 Vectorized Seeds
> |Architecture|16|32|64|128|256|512|1024|2048|4096|
> |--|--|--|--|---|---|---|----|----|----|
> |S5|1.00±0.00|1.00±0.00|1.00±0.00|1.00±0.00|1.00±0.00|1.00±0.00|0.50±0.23|0.00±0.19|0.25±0.16|
> |Transformer|1.00±0.00|0.75±0.22|1.00±0.00|1.00±0.00|1.00±0.00|0.70±0.27|-0.25±0.07|0.05±0.19|0.15±0.15|
> |LSTM|0.85±0.09|0.50±0.27|0.10±0.21|0.20±0.22|0.35±0.18|0.25±0.25|-0.05±0.08|0.10±0.15|0.15±0.15|
>
> We hope these latest results adequately address the concerns raised by the reviewers and would like to once again thank the reviewers for their time and consideration.

---

### Decision · Program_Chairs · 2023-09-21

**Decision:**

Accept (poster)

**Comment:**

This paper develops a modified version of S5 architecture as a drop-in replacement of RNNs and Transformers in POMDP and memory-intensive RL tasks, which improves the long-range sequence modeling and inference cost. All the reviewers appreciate the significance of the problem and the contribution/novelty of the paper. And the questions raised by the reviewers have been well addressed during the rebuttal period.